# OpenReview forum: "EchoMind: An Interrelated Multi-level Benchmark for Evaluating Empathetic Speech Language Models"
_ICLR.cc/2026/Conference — ICLR 2026 Poster_

### Official Review · Reviewer_Ywxa · 2025-10-29

**Soundness:** 3
**Presentation:** 2
**Contribution:** 2
**Rating:** 4
**Confidence:** 4

**Summary:**

The paper introduces EchoMind, a benchmark designed to evaluate the empathetic capabilities of Speech Language Models (SLMs) in dialogue, addressing the limitation of existing benchmarks that test linguistic, acoustic, and reasoning abilities in isolation. The benchmark employs an empathy-oriented framework including 39 vocal attributes, and its evaluation of 12 advanced SLMs reveals that almost all models struggle with highly expressive vocal cues, restricting their ability to generate contextually and emotionally aligned empathetic responses.

**Strengths:**

The paper introduces EchoMind, a large-scale benchmark designed to evaluate empathy in Speech Language Models (SLMs) through tasks of understanding, reasoning, and conversation. It initiates a systematic direction for assessing empathetic capability in SLMs, which is largely overlooked in prior speech or multimodal benchmarks. The authors back this up with extensive experiments on 12 advanced SLMs, and provide comprehensive analyses covering prompt sensitivity, synthetic vs. human speech robustness, and upper-bound performance under ideal vocal cue recognition.

**Weaknesses:**

See questions.

**Questions:**

1. The approach to synthesizing *hoarse* voices via voice cloning seems questionable, as such vocal characteristics are typically physiological rather than stylistic. Similarly, generating environmental-sound speech by simply mixing clean TTS outputs with background noise may not accurately reflect real-world acoustic conditions. Could the authors clarify how these methods ensure realism and avoid artifacts that might bias evaluation results?
2. Using vocabulary-level metrics such as BLEU or ROUGE to compare model responses against gold references does not seem to capture empathetic ability, since model generates responses in a free-form way. Could the authors justify the inclusion of these metrics or explain how they relate to empathy-oriented evaluation?
3. Table 5 shows large discrepancies between human and model-as-a-judge scores, particularly for GPT-4o-Audio, which raises concerns about the reliability of automatic judgments. Could the authors provide more details on the size of the sampled subset and discuss why such divergences occur?
4. The abbreviations *C1–C4* and *P1–P3* are difficult to interpret without repeatedly referring back to the text. Providing clearer, descriptive names would make the evaluation setup easier to follow.

---

> ### Author Response · Authors · 2025-11-21
> **Replies to Reviewer Ywxa——Weakness 1&2&3**
>
> Thank you for your time, careful review of our paper. Here are our detailed responses:
>
> >### **W1**. Concerns about synthesizing hoarse voices via voice cloning and generating environmental-sound speech by mixing clean TTS outputs with background noise.
>
> We fully understand your concern. In designing our voice synthesis approach, we categorize voice attributes into three types:
>  - **Global style** attributes: characteristics that remain consistent throughout the entire speech, such as gender or age.
>  - **Local style** attributes: features that appear only in specific segments, such as coughing.
>  - **Hybrid** attributes: a fusion of both global and local features. For example, “happy” emotion may originate from localized vocal cues, but its perceptual judgment still relies on the overall style.
>
> While hoarseness is generally physiological, we regard it as a global style attribute because it remains consistent across all phonemes in speech. Consequently, we consider voice cloning a suitable method for synthesizing hoarse voices.
>
> For environmental-sound speech, due to resource constraints, we followed prior work [1][2] and generated it by mixing clean TTS outputs with background noise.
>
> _______
>
> >### **W2**. Concerned that vocabulary-level metrics such as BLEU or ROUGE may not capture empathetic ability in free-form responses, please justify the inclusion of these metrics and explain how they relate to empathy-oriented evaluation?
>
> The ground-truth responses we provide are generated reference responses that explicitly demonstrate empathetic ability. Therefore, calculating BLEU or ROUGE scores between these references and the SLM's free-form predictions can serve as an auxiliary means to assess the SLM's empathetic ability. In addition, our choice of such metrics is consistent with evaluation schemes used in prior related work [1][2].
>
> _______
>
> >### **W3**. Concerned about discrepancies between human and model-as-a-judge scores, especially for GPT‑4o‑Audio. Could the authors provide more details on the size of the sampled subset and discuss why such divergences occur?
>
> The sampled subset size is 137, evaluated by three evaluators across several fine-grained dimensions for three models (**see human evaluation platform in Appendix B.4: Figure 7 & Figure 8** )
>
> We previously explained this misalignment in our submitted manuscript, which now appears in Lines 478–483 of the rebuttal version
> Specifically, the discrepancies primarily arise in Text-C2 and Audio‑VES, with **GPT-4o-Audio receiving notably lower human scores, this is mainly because evaluators found its responses overly long and formally structured, and its synthesized voice more formal in timbre, whereas other models sounded softer and warmer — traits linked to higher perceived empathy.**
>
> Additionally, we conducted supplementary assessments using Gemini‑2.5‑Pro for metrics C1–C4. we use Gemini‑2.5‑Pro for supplementary assessment of metrics C1–C4. As shown in Table A below, its evaluations correlate with GPT‑4o’s results at 0.90, 0.85, 0.81, and 0.64 for the four dimensions, respectively. This indicates strong overall alignment, with C4 (speech information relevance) displaying slightly weaker but still positive correlation.
> _**We present this supplementary evaluation in Lines 417–422 of the main text, and include the complete experimental results in Appendix C.1.**_
>
> **Table A**. Evaluation performance of Gemini‑2.5‑Pro on metrics C1–C4
> | Model                 | C1   | C2   | C3   | C4   |
> |-----------------------|------|------|------|------|
> | Audio-Flamingo3       |  1.1 | 1.04 | 1.44 | 2.09 |
> | Audio-Flamingo3+Think | 1.07 | 1.03 | 1.08 | 2.83 |
> | Audio-Flamingo3-chat  | 3.27 | 2.54 | 2.46 | 2.16 |
> | DeSTA2.5-Audio        |  4.2 | 3.81 | 3.47 | 2.68 |
> | VITA-Audio            | 4.49 | 4.19 | 4.46 | 2.89 |
> | LLaMA-Omni2           | 4.41 | 3.88 |  3.4 |  2.1 |
> | Baichuan-Omni-1.5     |  4.6 | 3.87 | 2.93 | 1.03 |
> | GLM-4-voice           | 4.09 |    4 | 4.09 | 2.28 |
> | OpenS2S               | 4.39 | 3.99 | 3.86 | 2.96 |
> | Qwen2.5-Omni-7B       | 4.02 | 3.94 | 4.29 | 2.39 |
> | Kimi-Audio            | 3.79 |  3.8 | 4.41 | 3.18 |
> | Step-Audio            | 4.49 | 4.37 | 4.24 | 2.45 |
> | EchoX                 | 2.99 | 2.12 | 2.48 |  1.7 |
> | GPT-4o-Audio          | 4.71 | 4.32 | 3.97 | 3.01 |

---

> ### Author Response · Authors · 2025-11-24
> **Replies to Reviewer Ywxa——Weakness 4**
>
> >### **W4**. Recommends using clearer, descriptive names (C1–C4 and P1–P3) to improve readability of the evaluation setup.
>
> Thank you for the suggestion. We will revise these abbreviations as follows to make them easier to follow.
> **(Please note that in the revised version for this rebuttal, we have temporarily retained the original abbreviations to avoid possible confusion for other reviewers. In the final CR version, we will unify them to the expressions below.)**
>  - C1 (Context Fit) → $\text{C}_\text{CtxFit}$
>  - C2 (Response Naturalness) → $\text{C}_\text{RespNat}$
>  - C3 (Colloquialism Degree) → $\text{C}_\text{ColloqDeg}$
>  - C4 (Speech Information Relevance) → $\text{C}_\text{SpeechRel}$
>  - P1 (Zero Prompt) → $\text{P}_\text{Zero}$
>  - P2 (Basic Prompt) → $\text{P}_\text{Basic}$
>  - P3 (Enhanced Prompt) → $\text{P}_\text{Enhance}$
>
>
>
> ### Reference
> [1] Sd-eval: A benchmark dataset for spoken dialogue understanding beyond words. NeurIPS 2024
> [2] VoxDialogue: Can Spoken Dialogue Systems Understand Information Beyond Words? ICLR 2025

---

> > ### Author Response · Authors · 2025-11-27
> > **Revised Manuscript and Addressed Reviewer Concerns**
> >
> > Dear Reviewer Ywxa,
> >
> > We have provided a detailed response to all of your questions and suggestions, and have uploaded a revised manuscript accordingly.
> >
> > **If our responses have addressed your concerns, we would kindly appreciate it if you could consider re-evaluating our score.**
> >
> > Thank you very much for your time and consideration.
> >
> > Best,
> > Authors of Submission15900

---

### Official Review · Reviewer_eEYg · 2025-10-30

**Soundness:** 2
**Presentation:** 3
**Contribution:** 3
**Rating:** 4
**Confidence:** 4

**Summary:**

This paper introduces EchoMind, a benchmark designed to evaluate the empathetic capabilities of Speech Language Models (SLMs) through a multi-level framework consisting of understanding, reasoning, and conversation tasks. The benchmark uses controlled vocal-style variations of semantically neutral scripts across 39 vocal attributes spanning speaker information, paralinguistic cues, and environmental sounds. The authors evaluate 12 advanced SLMs and find that even state-of-the-art models struggle with highly expressive vocal cues, particularly in generating empathetically aligned responses. While the paper addresses an important gap in evaluating emotional intelligence in SLMs, it suffers from several methodological limitations and presentation issues that prevent acceptance at this time.

**Strengths:**

- The focus on empathetic capabilities in SLMs addresses a critical gap in current benchmarks, moving beyond pure linguistic understanding to emotional intelligence.
- The benchmark uses controlled vocal-style variations of semantically neutral scripts across 39 vocal attributes spanning speaker information, paralinguistic cues, and environmental sounds.

**Weaknesses:**

- The study introduces multi fine-grained tasks, but results are only reported at the coarse levels (understanding, reasoning, conversation). Showing results for each sub-task would reveal which empathy aspects, like tone recognition or emotional adaptation, remain most challenging.

- While the paper lists 39 vocal attributes, their individual contributions to empathy are not analyzed. Exploring which features (e.g., pitch, tempo, timbre) drive performance would make the work more insightful and interpretable.

- The human–TTS comparison is interesting but stops at showing a performance gap. It’s unclear whether the issue comes from model robustness or unrealistic TTS cues(39 vocal attributes is reasonable?). Diagnostic experiments could clarify this.

- The human evaluation is on too small a sample (6 cases per vocal-cue type for 3 models) to draw strong conclusions.Expanding the dataset and adding inter-rater agreement (e.g., Cohen’s Kappa) would strengthen the conclusions.

- GPT-4o is used both to generate  “ground truth” and as a model under evaluation. This creates circular bias and may inflate its scores. An independent labeling model is recommended.

-  Typos: Line 291  two ”, ,” ; Line 263: "shown as Table 2"-->"as shown in Table 2"

**Questions:**

- What are the exact prompts used for GPT-4o to generate scripts and MCQs?
- How do you justify calling this an "empathy" benchmark when it primarily tests acoustic feature detection?
- Why not include more analysis of which specific vocal attributes are most challenging?

---

> ### Author Response · Authors · 2025-11-21
> **Replies to Reviewer eEYg——Weakness 1 (part 1)**
>
> We thank the reviewer for the thorough evaluation, constructive suggestions, and valuable comments. **We have incorporated all relevant analyses and experiments into the revised manuscript**, and provide detailed responses below.
>
> >### **W1**. Suggest showing results for each sub-task would reveal which empathy aspects, like tone recognition or emotional adaptation, remain most challenging.
>
> Thank you for your suggestion. A more detailed analysis will indeed enrich the value of our work.
>
> We perform a fine-grained analysis of the model's performance across 8 sub-tasks in voice understanding and 10 sub-tasks in reasoning tasks, with the results shown in Table A and Table B below. The findings indicate that _Voice Style Detection_, which requires interpreting physiological states from audio, is among the most difficult tasks, alongside _Background Sound Detection_. Performance is also notably lower on the two reasoning tasks, _Preceding Event Inference_ and _Empathy‑Aware Response Selection_, underscoring the difficulty SLMs face in generating emotionally intelligent responses based on vocal cues.
>
>
> _**We present this fine‑grained analysis in Lines 421–428 of the main text, and include the complete experimental results in Appendix C.2.**_
>
>
>
>  **Table A:** Model performance across 8 sub-tasks in voice understanding MCQ task.
> | Model                 | Coarse | Gender | Age Group | Voice Style | Speech Emotion | Speaking Pace |   NVE | Background |
> |-----------------------|-------:|-------:|----------:|------------:|---------------:|--------------:|------:|-----------:|
> | Audio-Flamingo3       |  58.43 |  77.27 |     54.17 |       60.34 |          45.97 |         74.31 | 77.08 |      48.73 |
> | Audio-Flamingo3+Think |  37.01 |  50.91 |     30.21 |       33.91 |          25.31 |         46.53 | 47.92 |      27.71 |
> | Audio-Flamingo3-chat  |  58.45 |  55.45 |     57.29 |       60.06 |          38.41 |         56.94 | 73.51 |      54.78 |
> | DeSTA2.5-Audio        |  61.42 |  88.18 |     50.52 |       42.53 |          54.28 |         57.64 | 69.35 |      38.54 |
> | VITA-Audio            |  25.66 |  50.91 |     30.21 |       21.55 |          21.91 |         47.92 | 24.40 |      24.84 |
> | LLaMA-Omni2           |  33.62 |  50.00 |     29.17 |       33.33 |          43.32 |         42.36 | 29.76 |      27.71 |
> | Baichuan-Omni-1.5     |  42.95 |  54.63 |     28.65 |       31.90 |          53.60 |         51.41 | 57.01 |      33.56 |
> | GLM-4-voice           |  23.59 |  46.08 |     27.78 |       26.12 |          25.35 |         51.69 | 27.52 |      26.80 |
> | OpenS2S               |  30.33 |  53.64 |     57.29 |       23.28 |          28.97 |         47.92 | 39.58 |      28.34 |
> | Qwen2.5-Omni-7B       |  58.84 |  72.73 |     56.25 |       57.76 |          56.87 |         51.39 | 67.26 |      55.41 |
> | Kimi-Audio            |  53.40 |  55.66 |     35.16 |       37.72 |          39.61 |         54.10 | 54.49 |      43.39 |
> | Step-Audio            |  48.14 |  51.82 |     33.85 |       32.76 |          51.89 |         65.28 | 50.00 |      26.52 |
> | EchoX                 |  26.23 |  47.27 |     26.70 |       21.84 |          22.42 |         45.77 | 24.40 |      24.92 |
> | GPT-4o-Audio          |  65.15 |  72.34 |     44.26 |       55.91 |          67.81 |         75.00 | 79.40 |      46.15 |
> | Average               |  44.52 |  59.06 |     40.11 |       38.50 |          41.12 |         54.88 | 51.55 |      36.24 |

---

> ### Author Response · Authors · 2025-11-21
> **Replies to Reviewer eEYg——Weakness 1 (part 2)**
>
> **Table B:** Model performance across 10 sub-tasks in reasoning MCQ task.
> | Model                 | Multi-People | Laughter-SenTM | Shouting-SenTM | Audio-Text-SenTM | Response-ST | Personalized-Rec | Contextual-SugGT | Preceding-Event | Speaker-Intent | Empathy-Aware-Res |
> |-----------------------|--------------|----------------|----------------|------------------|-------------|------------------|------------------|-----------------|----------------|-------------------|
> | Audio-Flamingo3       |        55.24 |          44.83 |          62.50 |            24.24 |       58.70 |            80.00 |            82.22 |           52.88 |          70.27 |             38.54 |
> | Audio-Flamingo3+Think |        50.40 |          44.83 |          62.50 |            14.14 |       55.77 |            39.18 |            77.33 |           30.89 |          45.71 |             30.09 |
> | Audio-Flamingo3-chat  |        54.62 |          44.83 |          34.38 |            85.86 |       49.86 |            62.54 |            78.22 |           50.38 |          58.38 |             32.36 |
> | DeSTA2.5-Audio        |        50.40 |          41.38 |          56.25 |            29.29 |       65.49 |            79.84 |            82.44 |           48.37 |          73.24 |             50.97 |
> | VITA-Audio            |        50.00 |          44.83 |          62.50 |            85.86 |       27.72 |            28.25 |            27.33 |           25.06 |          35.68 |             24.69 |
> | LLaMA-Omni2           |        47.98 |          48.28 |          65.63 |            15.15 |       50.00 |            64.76 |            78.89 |           48.87 |          62.43 |             31.31 |
> | Baichuan-Omni-1.5     |        49.12 |          58.62 |          56.25 |            58.62 |       56.25 |            71.78 |            76.71 |           44.76 |          65.85 |             38.24 |
> | GLM-4-voice           |        51.02 |          53.85 |          46.67 |            24.36 |       25.08 |            24.21 |            26.45 |           24.52 |          24.75 |             27.02 |
> | OpenS2S               |        50.00 |          44.83 |          62.50 |            27.27 |       60.33 |            20.67 |            74.89 |           45.86 |          58.38 |             49.29 |
> | Qwen2.5-Omni-7B       |        50.40 |          44.83 |          53.13 |            41.41 |       63.04 |            69.90 |            81.78 |           46.62 |          62.43 |             43.65 |
> | Kimi-Audio            |        60.48 |          51.72 |          37.50 |            44.57 |       58.50 |            70.28 |            81.56 |           48.48 |          64.21 |             35.50 |
> | Step-Audio            |        51.61 |          51.72 |          40.63 |            49.49 |       61.41 |            60.63 |            79.11 |           45.11 |          62.16 |             45.59 |
> | EchoX                 |        50.00 |          44.83 |          62.50 |            69.86 |       28.69 |            46.19 |            38.44 |           33.76 |          40.82 |             24.67 |
> | GPT-4o-Audio          |        61.38 |          44.83 |          59.38 |            46.81 |       69.57 |            78.73 |            87.33 |           59.30 |          74.86 |             58.64 |
> | Average               |        52.33 |          47.44 |          54.45 |            44.07 |       52.17 |            56.93 |            69.48 |           43.20 |          57.08 |             37.90 |

---

> ### Author Response · Authors · 2025-11-21
> **Replies to Reviewer eEYg——Weakness 2 and Question 3 (Part 1)**
>
> >### **W2 & Q3**. Suggest exploring which features (e.g., pitch, tempo, timbre) drive performance would make the work more insightful and interpretable.
>
> Thanks for the great suggestion. **Through feature analysis, we find that SLMs exhibit increased sensitivity to high‑pitch audio inputs, resulting in enhanced performance across tasks at all three levels. By contrast, speaker gender, whether male or female, has negligible impact on model performance.** The specific analysis method is as follows:
>  - **Impact of audio pitch**: We calculate the pitch values of 1,137 target expression audio inputs, classifying the top 300 pitch values as the high-pitch group and those below 300 as the low-pitch group. We then filter the related task case results and evaluate the outcomes. We evaluate this impact using the accuracy (ACC) for voice understanding and reasoning tasks, and C4 (speech information relevance) and VES (Vocal Empathy Score) for conversation responses. **The results in Table C SLMs exhibit increased sensitivity to high‑pitch audio inputs, resulting in enhanced performance across tasks at all three levels.**
>  - **Impact of voice gender**: We divide the target expression audio inputs into male and female voice groups (Line 234-235: ensuring even gender representation across synthesis conditions to avoid bias). **Table D shows that while there are slight differences in voice timbre across SLMs based on gender, the overall impact on model performance is minimal**.
>
> _**This feature analysis is described in Lines 421–428 of the main text, with the full experimental results available in Appendix C.3 and C.4.**_
>
>
>
> **Table C:** Impact of audio pitch on SLM performance in voice understanding, reasoning, and conversation tasks
> | Model                 | Understanding (high) | Understanding (low) | Reasoning (high) | Reasoning (low) | C4 (high) | C4 (low) | VES (high) | VES (low) |
> |-----------------------|----------------------|---------------------|------------------|-----------------|-----------|----------|------------|-----------|
> | Audio-Flamingo3       | 65.83                | 63.33               | 59.84            | 54.19           | 2.02      | 1.95     | -          | -         |
> | Audio-Flamingo3+Think | 65.11                | 67.50               | 42.63            | 39.76           | 2.67      | 2.47     | -          | -         |
> | Audio-Flamingo3-chat  | 42.00                | 40.50               | 54.61            | 45.27           | 3.08      | 2.97     | -          | -         |
> | DeSTA2.5-Audio        | 62.33                | 53.50               | 64.55            | 58.49           | 3.7       | 3.19     | -          | -         |
> | VITA-Audio            | 25.33                | 26.83               | 28.28            | 30.65           | 3.3       | 2.86     | 2.14       | 2.1       |
> | LLaMA-Omni2           | 36.50                | 37.83               | 51.43            | 46.56           | 3.06      | 2.87     | 2.05       | 2.17      |
> | Baichuan-Omni-1.5     | 46.15                | 44.27               | 57.07            | 51.71           | 3.11      | 2.72     | 2.44       | 2.36      |
> | GLM-4-voice           | 25.37                | 25.73               | 25.51            | 29.40           | 3.16      | 2.88     | 3.01       | 2.87      |
> | OpenS2S               | 35.83                | 29.33               | 51.35            | 46.26           | 3.65      | 3.1      | 3.11       | 2.77      |
> | Qwen2.5-Omni-7B       | 68.17                | 60.10               | 58.50            | 54.31           | 3.04      | 2.74     | 3.27       | 3.26      |
> | Kimi-Audio            | 46.78                | 51.77               | 58.75            | 51.89           | 3.46      | 3.26     | 3.13       | 2.79      |
> | Step-Audio            | 40.83                | 44.00               | 57.68            | 55.81           | 3.39      | 2.93     | 3.25       | 3.18      |
> | EchoX                 | 25.75                | 26.92               | 34.41            | 32.76           | 2.27      | 2.14     | 1.4        | 1.44      |
> | GPT-4o-Audio          | 70.78                | 66.33               | 68.38            | 65.86           | 3.64      | 3.35     | 3.15       | 3.55      |
> | Average               | 46.91                | 45.57               | 50.93            | 47.35           | 3.11      | 2.82     | 2.70       | 2.65      |

---

> > ### Author Response · Authors · 2025-11-24
> > **Replies to Reviewer eEYg——Weakness 2 and Question 3 (Part 2)**
> >
> > **Table D:** Impact of voice gender (male vs. female) on SLM performance.
> > | Model                 | Understanding (male) | Understanding (female) | Reasoning (male) | Reasoning (female) | C4 (male) | C4 (female) | VES (male) | VES (female) |
> > |-----------------------|----------------------|------------------------|------------------|--------------------|-----------|-------------|------------|--------------|
> > | Audio-Flamingo3       |                63.84 |                  64.76 |            56.56 |              61.17 |      2.02 |        1.92 | -          | -            |
> > | Audio-Flamingo3+Think |                65.74 |                  64.55 |            41.53 |              44.45 |      2.49 |        2.59 | -          | -            |
> > | Audio-Flamingo3-chat  |                41.00 |                  41.41 |            50.05 |              53.21 |      2.98 |        2.99 | -          | -            |
> > | DeSTA2.5-Audio        |                56.06 |                  57.33 |            60.71 |              65.49 |      3.38 |        3.33 | -          | -            |
> > | VITA-Audio            |                25.26 |                  25.22 |            30.11 |              29.28 |      3.02 |        3.04 |       2.17 |         2.08 |
> > | LLaMA-Omni2           |                36.51 |                  35.96 |            49.81 |              51.39 |      2.92 |         2.9 |       2.06 |         2.07 |
> > | Baichuan-Omni-1.5     |                41.65 |                  45.57 |            54.80 |              56.24 |      2.87 |        2.97 |       2.46 |         2.34 |
> > | GLM-4-voice           |                26.16 |                  24.92 |            27.13 |              26.35 |      2.89 |        2.96 |       2.98 |         2.89 |
> > | OpenS2S               |                31.31 |                  31.04 |            48.59 |              52.26 |      3.23 |         3.4 |       2.77 |         3.06 |
> > | Qwen2.5-Omni-7B       |                61.70 |                  60.02 |            56.35 |              59.12 |      2.83 |        2.86 |       3.26 |         3.22 |
> > | Kimi-Audio            |                48.47 |                  50.09 |            54.89 |              57.00 |       3.3 |        3.37 |       2.79 |         2.98 |
> > | Step-Audio            |                40.40 |                  41.09 |            55.53 |              56.17 |      3.08 |         3.1 |       3.21 |         3.19 |
> > | EchoX                 |                25.89 |                  25.54 |            34.83 |              35.44 |       2.2 |        2.19 |       1.41 |         1.39 |
> > | GPT-4o-Audio          |                66.87 |                  65.62 |            65.97 |              70.19 |      3.47 |         3.4 |       3.42 |         3.26 |
> > | Average               |                45.06 |                  45.22 |            49.06 |              51.27 |     2.917 |        2.93 |       2.65 |         2.65 |

---

> > > ### Author Response · Authors · 2025-11-24
> > > **Replies to Reviewer eEYg——Weakness 3&4**
> > >
> > > >### **W3**. Suggest conducting diagnostic experiments to explore the reasons behind the performance differences between human and TTS comparisons.
> > >
> > > Thank you for your suggestion. First, we emphasize that during the human-TTS comparison, we strictly controlled the variables by using identical scripts, with the same gender and consistent voice attributes for each script. In addition, all recordings were conducted in a soundproof studio to eliminate potential environmental noise. This ensures any performance differences are due to the source (human vs. synthetic).
> > >
> > > Additionally, we grouped five fine-grained paralinguistic dimensions and compared the human-TTS performance differences, as shown in the Table E below. **The average performance differences are: NVE > Physiological State > Speed > Emotion > Volume. This pattern likely reflects the relative difficulty of synthesizing these attributes for TTS; more challenging attributes result in greater performance differences, possibly because they are less represented in the SLM's training corpus.**
> > >
> > > _**We include this voice group analysis in Appendix B.4.**_
> > >
> > > **Table E:** Comparison of Δ(human-TTS) performance differences across five fine-grained paralinguistic dimensions.
> > > |                     | C1    | C2    | C3    | C4    | VES   | Average |
> > > |---------------------|-------|-------|-------|-------|-------|---------|
> > > | Physiological State | -0.31 | -0.25 | -0.19 | -0.16 |  0.01 |   -0.18 |
> > > | Emotion             | -0.22 | -0.06 | -0.11 | -0.10 |  0.04 |   -0.09 |
> > > | Volume              | -0.02 |  0.03 |     0 | -0.03 | -0.29 |   -0.06 |
> > > | Speed               | -0.15 | -0.16 | -0.11 | -0.15 | -0.18 |   -0.15 |
> > > | NVE                 | -0.40 | -0.30 | -0.33 | -0.35 | -0.39 |   -0.35 |
> > >
> > > _______
> > >
> > > >### **W4**. Concerned that the human evaluation is based on too small a sample size (6 cases per vocal-cue type for 3 models).
> > >
> > > We fully understand your concern. Although the sample size per vocal-cue type is only 6, it is important to emphasize that we use multiple vocal-cue settings, resulting in a total of 137 audio input cases finally. These cases are evaluated by three evaluators across several fine-grained dimensions for three models, as shown in Figures 7 and 8 in Appendix B.3. This makes the human evaluation process quite extensive. Based on feedback from the evaluators, the assessment for Figure 7 (Fine-grained assessments of three models across six dimensions) takes an average of 7 hours per evaluator, while the evaluation for Figure 8 (Evaluation of the differences in model responses to the same script delivered in different vocal styles) takes approximately 1.5 hours per evaluator.
> > >
> > > In addition, we analyze inter‑rater agreement across six evaluation dimensions and three models, based on ratings from three evaluators (Table F). Agreement varies across dimensions and models, with Qwen2.5‑Omni‑7B showing the highest consistency and GPT‑4o‑Audio the lowest. This inverse relationship between model performance and agreement aligns with Popper’s falsifiability principle: errors are generally easier to identify than correctness. Low‑performing models tend to produce clear, recognizable faults, facilitating consensus; high‑performing models yield outputs largely free of obvious errors, where subtle quality differences invite subjective interpretations, reducing agreement.
> > >
> > > To mitigate individual bias, we average the scores from three evaluators to obtain the final human rating for each model (Table 5). We then compare these averages with Model‑as‑Judge results (Table G), finding moderate overall consistency—particularly lower in dimensions tied to auditory cues (Text‑C4 and Audio‑VES)—underscoring the inherent challenges of this subjective evaluation.
> > >
> > >
> > > _**We include this analysis in Appendix B.4.**_
> > >
> > > **Table F:** Inter-rater agreement across metrics/models **among three elevators**
> > > | Model           | Text-C1 | Text-C2 | Text-C3 | Text-C4 | Audio-VES | Audio-Quality | Average |
> > > |-----------------|---------|---------|---------|---------|-----------|---------------|---------|
> > > | Qwen2.5-Omni-7B | 0.59    | 0.82    | 0.76    | 0.58    | 0.51      | 0.54          |    0.63 |
> > > | Step-Audio      | 0.54    | 0.60    | 0.58    | 0.60    | 0.53      | 0.67          |    0.59 |
> > > | GPT-4o-Audio    | 0.51    | 0.48    | 0.52    | 0.53    | 0.38      | 0.44          |    0.48 |
> > >
> > > **Table G:** Inter-rater agreement across metrics/models **between Model-as-Judge and human evaluation.**
> > > | Model           | Text-C1 | Text-C2 | Text-C3 | Text-C4 | Audio-VES | Audio-Quality | Average |
> > > |-----------------|---------|---------|---------|---------|-----------|---------------|---------|
> > > | Qwen2.5-Omni-7B | 0.65    | 0.77    | 0.80    | 0.49    | 0.39      | 0.81          |    0.65 |
> > > | Step-Audio      | 0.66    | 0.51    | 0.45    | 0.55    | 0.45      | 0.72          |    0.56 |
> > > | GPT-4o-Audio    | 0.47    | 0.55    | 0.53    | 0.45    | 0.34      | 0.74          |    0.51 |

---

> > > > ### Author Response · Authors · 2025-11-24
> > > > **Replies to Reviewer eEYg——Weakness 5&6 and Question 1&2**
> > > >
> > > > >### **W5**.GPT-4o is used both to generate "ground truth" and as a model under evaluation. This creates circular bias and may inflate its scores. An independent labeling model is recommended.
> > > >
> > > > For clarity, GPT‑4o evaluates model responses without referencing the generated "ground truth". Evaluation is based solely on user‑level input scripts and voice attributes across multiple dimensions.
> > > >
> > > > To mitigate potential bias when GPT‑4o also evaluates GPT‑4o‑Audio outputs, we use Gemini‑2.5‑Pro to score metrics C1–C4. As shown in Table F below, its evaluations for GPT‑4o‑Audio largely align with GPT‑4o's: C1 ranked first, C2 and C4 second, and C3 did not place in the top tier.
> > > > The spearman correlation coefficients between the two model‑based evaluators were 0.90, 0.85, 0.81, and 0.64 for C1 through C4, respectively, indicating strong agreement across most metrics.
> > > > The lower correlation for C4 (speech information relevance) likely reflects inherent challenges in evaluating nuanced auditory cues.
> > > >
> > > > _**We present this supplementary evaluation in Lines 417–422 of the main text, and include the complete experimental results in Appendix C.1.**_
> > > >
> > > >
> > > >
> > > > **Table F**. Evaluation performance of Gemini‑2.5‑Pro on metrics C1–C4
> > > > | Model                 | C1   | C2   | C3   | C4   |
> > > > |-----------------------|------|------|------|------|
> > > > | Audio-Flamingo3       |  1.1 | 1.04 | 1.44 | 2.09 |
> > > > | Audio-Flamingo3+Think | 1.07 | 1.03 | 1.08 | 2.83 |
> > > > | Audio-Flamingo3-chat  | 3.27 | 2.54 | 2.46 | 2.16 |
> > > > | DeSTA2.5-Audio        |  4.2 | 3.81 | 3.47 | 2.68 |
> > > > | VITA-Audio            | 4.49 | 4.19 | 4.46 | 2.89 |
> > > > | LLaMA-Omni2           | 4.41 | 3.88 |  3.4 |  2.1 |
> > > > | Baichuan-Omni-1.5     |  4.6 | 3.87 | 2.93 | 1.03 |
> > > > | GLM-4-voice           | 4.09 |    4 | 4.09 | 2.28 |
> > > > | OpenS2S               | 4.39 | 3.99 | 3.86 | 2.96 |
> > > > | Qwen2.5-Omni-7B       | 4.02 | 3.94 | 4.29 | 2.39 |
> > > > | Kimi-Audio            | 3.79 |  3.8 | 4.41 | 3.18 |
> > > > | Step-Audio            | 4.49 | 4.37 | 4.24 | 2.45 |
> > > > | EchoX                 | 2.99 | 2.12 | 2.48 |  1.7 |
> > > > | GPT-4o-Audio          | 4.71 | 4.32 | 3.97 | 3.01 |
> > > >
> > > > _______
> > > >
> > > > >### **W6**. Two Typos
> > > >
> > > > Thank you for the careful review. We have made the changes in the revised version.
> > > >
> > > > _______
> > > >
> > > > >### **Q1**: What are the exact prompts used for GPT-4o to generate scripts and MCQs?
> > > >
> > > > We use GPT‑4o to generate dialogue scripts, but apply heuristic rules to construct the corresponding MCQs (Appendix A.4 provides details of these rules) to ensure that each question has a single, clearly correct answer.
> > > >
> > > > The use of GPT‑4o to generate scripts is not a one‑shot process. There are subtle differences across various coarse‑grained vocal dimensions. Moreover, to obtain the final meaningful scripts, this process involves multiple iterations of generation and verification. **The complete workflow is illustrated in Appendix B.1(Figure 5), and the initial script‑generation prompt is provided in the supplementary materials.**
> > > >
> > > > _______
> > > >
> > > > >### **Q2**: How do you justify calling this an "empathy" benchmark when it primarily tests acoustic feature detection?
> > > >
> > > > Our ultimate goal is empathetic dialogue, while testing acoustic feature detection serves as an evaluation of low‑level abilities.

---

> > > > > ### Author Response · Authors · 2025-11-27
> > > > > **Revised Manuscript and Addressed Reviewer Concerns**
> > > > >
> > > > > Dear Reviewer eEYg,
> > > > >
> > > > > We have provided a detailed response to all of your questions and suggestions, and have uploaded a revised manuscript accordingly.
> > > > >
> > > > > **If our responses have addressed your concerns, we would kindly appreciate it if you could consider re-evaluating our score.**
> > > > >
> > > > > Thank you very much for your time and consideration.
> > > > >
> > > > > Best,
> > > > > Authors of Submission15900

---

### Official Review · Reviewer_Da1E · 2025-11-01

**Soundness:** 2
**Presentation:** 3
**Contribution:** 2
**Rating:** 4
**Confidence:** 4

**Summary:**

The paper presents EchoMind, the first interrelated, multi-level benchmark designed to evaluate the empathetic and socially intelligent capabilities of Speech Language Models (SLMs). It aims to integrate understanding, reasoning, and conversational response tasks grounded in vocal cues, testing models across 39 vocal attributes. The benchmark includes both synthetic and limited human-recorded data and evaluates 12 SLMs using objective and subjective metrics.

**Strengths:**

This is the first benchmark explicitly targeting speech-based social intelligence and empathetic understanding. The effort to formalize empathy-related evaluation dimensions in SLMs is both timely and valuable for the community.

**Weaknesses:**

- While the paper motivates the work by criticising existing benchmarks for evaluating capabilities in isolation, the proposed benchmark does not actually integrate these skills in a meaningful way. The three “levels” (understanding, reasoning, and conversation) are still evaluated independently, using separate metrics. What is presented feels like more dimensions, not genuine integration. A stronger demonstration would involve an Arena-style extrinsic evaluation assessing the real downstream impact of empathy-aware dialogue systems.
- The reported high inter-dimensional correlations undermine the paper’s central claim about interrelated evaluation. If the metrics are highly correlated, this suggests that the tasks may not capture distinct or complementary aspects of empathetic ability — further challenging the notion of “multi-level” integration.
- The benchmark relies heavily on synthetic data generated by LLMs and TTS systems, with only two human speakers contributing real recordings—and both are non-native English speakers. This raises serious concerns about data diversity, realism, and ecological validity. Additionally, although the paper states that synthetic data was “manually checked,” there is no evidence of systematic human evaluation or quality control beyond a few ad hoc examples.
- The human evaluation results are not statistically validated and, in fact, appear inconsistent with the model-based GPT-4o evaluations in half of the tests. It is, therefore, misleading to conclude “alignment” between human and model-based assessments. Without statistical tests or inter-rater reliability analysis, the claim lacks empirical support.

**Questions:**

See my comments above.

---

> ### Author Response · Authors · 2025-11-21
> **Replies to Reviewer  Da1E——Weakness 1&2**
>
> Thank you for your time, careful review of our paper.  **We have incorporated all suggested analyses and addressed all raised concerns in the revised manuscript.**
>
> Here are our detailed responses:
>
>
> >### **W1-1**: Criticize the benchmark for lacking  interrelated multi-level integration, as the three "levels" are evaluated independently;
> >### **W2**. Argue that the high correlations between task scores suggest the tasks may not capture distinct or complementary aspects of empathetic ability?
>
> We sincerely apologize for any misunderstanding arising from our used term "interrelated multi‑level" and its association with the "high correlations between tasks" result. We wish to clarify the following:
>
>
> - **Regarding W1‑1, EchoMind is explicitly designed as an interrelated multi‑level benchmark because all task levels use the exact same set of audio inputs and controlled vocal‑style variations.** While scores are reported independently for each level, this unified input design ensures a shared context across tasks, enabling correlation analyses between stages.  This approach is analogous to the benchmark design in CHARM [1], where reasoning and memorization tasks share a common input basis to allow direct correlation analysis. In contrast, most existing benchmarks (see Table 1) rely on disparate datasets across tasks, which can not reflect intrinsic inter‑task connections.
>
> - **Regarding W2, the high correlations between task scores are an intended and positive feature of the benchmark, not a limitation.** EchoMind's structure follows a hierarchical cognitive pipeline: perceptual understanding → integrated reasoning → empathetic dialogue, where downstream empathetic ability is theoretically dependent on upstream perceptual and reasoning skills. The observed correlations empirically validate this dependency, confirming that our multi‑level integration design is well‑founded. Moreover, the high correlation results provide a concrete roadmap: to achieve the highest‑quality empathetic dialogue, SLMs must first be improved in perceptual and reasoning capabilities.
>
> _**In the rebuttal revised manuscript (Lines 86–94), we provide a strengthened explanation of the concept of "interrelated multi‑level". Furthermore, in the Experimental Setup section, we explicitly emphasize its underlying prerequisite, the Audio Inputs Setup (Lines 351–353).**_
>
> _______
>
> >### **W1-2**:  Suggest an Arena-style extrinsic evaluation to assess the real downstream impact of empathy-aware dialogue systems.
>
> Thank you for the suggestion. In fact, when conducting the six-dimension evaluation of the three models (results shown in Table 5), we aimed to minimize evaluator subjectivity (e.g., occasional tendency toward over- or under-scoring) by presenting the outputs of all three models on the same page for comparative scoring (see human evaluation platform in Appendix B.4: Figure 7 & Figure 8). This setup inherently embodies certain characteristics of arena-style pairwise comparison.
>
> Accordingly, we conduct an Arena‑style evaluation, aggregating scores across six dimensions to produce an overall score for each model. Pairwise comparisons yield win/loss/tie counts and win rates (Table A below). Results reveal a ranking: GPT‑4o‑Audio > Step‑Audio > Qwen2.5‑Omni‑7B, which closely matches rankings from fine‑grained dimension‑by‑dimension scoring, confirming that aggregated scores validly represent overall performance. _**We include the arena‑style evaluation analysis in Lines 491–495 and Table 6 of the rebuttal revised manuscript.**_
>
> **Table A.** Arena-style Evaluation: Pairwise ranking of three models based on aggregated six‑dimension scores.
> | Model           | Win | Loss | Tie | Win Rate |
> |-----------------|-----|------|-----|----------|
> | Qwen2.5-Omni-7B | 232 | 349  | 241 | 0.28     |
> | Step-Audio      | 277 | 285  | 260 | 0.34     |
> | GPT-4o-Audio    | 346 | 221  | 255 | 0.42     |

---

> ### Author Response · Authors · 2025-11-21
> **Replies to Reviewer Da1E——Weakness 3**
>
> >### **W3**.Concerns about synthetic data from LLMs and TTS systems, alongside real recordings by non-native English human speakers, with manually verified details.
>
> To ensure data quality,
> - **All LLM‑generated utterances are verified by three annotators, and only those unanimously judged coherent and appropriate are retained (_mentioned in Lines 206–209_).** The selection criteria are: i) the synthetic utterance must be semantically neutral, without explicitly revealing any voice information; and ii) the intended meaning of the utterance must exhibit different interpretative tendencies when expressed in the target voice versus an alternative voice. _**The selection criteria are noted in Footnote 4 (Lines 214–215)**_.
> - **We clarify that all TTS‑generated audios containing paralinguistic cues are not produced in bulk with post‑hoc sampling inspection. Instead, each audio clip is synthesized manually in a one‑by‑one manner, with immediate quality checks during generation.** In some cases, dozens of synthesis attempts are required before obtaining an audio sample that meets the specified paralinguistic requirements.  **_We emphasize the manually verified details in Lines 229–233._**
>
> While the two human speakers in EchoMind are non‑native English speakers, **both have extensive immersion in English‑medium academic environments and are active members of a university voice‑acting society, with excellent English proficiency and professional voice‑acting skills.** This choice reflects common real‑world usage scenarios for speech language models, which often interact with proficient non‑native speakers, and represents our best effort within available resources. Moreover, **to ensure quality and consistency, each speaker recorded all scripts over three separate sessions of 2–3 hours, totaling ~7 hours for the male and ~7.5 hours for the female, covering all 39 vocal attributes with controlled delivery.**  _**We provide additional human recording details and audio recording platform in Appendix A.3.**_
>
> Furthermore, unlike prior work [2] using CosyVoice [3], which is limited in the range of vocal attributes it can produce, we adopt the more advanced GPT‑4o mini TTS along with other feasible synthesis approaches, enabling higher expressiveness and ensuring the generated audio meets high‑quality standards.

---

> ### Author Response · Authors · 2025-11-21
> **Replies to Reviewer Da1E—— Weakness 4**
>
> >### **W4**. The human evaluation results should be statistically validated, and the conclusion regarding alignment should be questioned.
>
> Thank you for highlighting the concern regarding the term "alignment". _**To avoid misunderstanding, we have changed the title to: "Human Evaluation — Partial Agreement with Model-based Automatic Metrics".**_
>
> Besides, we previously explained this misalignment in our submitted manuscript, which now appears in Lines 478–483 of the rebuttal version
> Specifically, in Text-C2 and Audio-VES, human evaluation results were not aligned with the Model-as-Judge outcomes, with **GPT-4o-Audio receiving notably lower human scores, this is mainly because evaluators found its responses overly long and formally structured, and its synthesized voice more formal in timbre, whereas other models sounded softer and warmer — traits linked to higher perceived empathy.**
>
>
>
> In addition, we analyze inter‑rater agreement across six evaluation dimensions and three models, based on ratings from three evaluators (Table B). Agreement levels vary across dimensions and models, reflecting differences in evaluators' perceptions. Qwen2.5‑Omni‑7B demonstrates the highest scoring consistency, whereas GPT‑4o‑Audio shows the lowest.
> The observed inverse relationship between model performance and inter‑rater agreement echoes Karl Popper’s falsifiability principle:errors are generally easier to demonstrate than correctness.
> Low‑performing models often produce outputs with clear, widely recognizable faults, enabling evaluators to reach consensus more easily. In contrast, high‑performing models generate outputs largely free of obvious errors, where quality differences are subtle and depend on subjective interpretation, reducing agreement.
>
>
> To reduce individual bias, we used the average score from the three evaluators as the final human rating for each model (see Table 5 in the paper). We then compared these average human scores with the Model-as-Judge evaluations (Table C below). Overall consistency is moderate, particularly lower in dimensions directly related to auditory cues (Text-C4 and Audio-VES), highlighting the challenges inherent in this subjective task.
>
> _**We include this analysis in Appendix B.4.**_
>
> **Table B:** Inter-rater agreement across metrics/models among three elevators
>   | Model           | Text-C1 | Text-C2 | Text-C3 | Text-C4 | Audio-VES | Audio-Quality | Average|
> |-----------------|---------|---------|---------|---------|-----------|---------------|---------|
> | Qwen2.5-Omni-7B | 0.59    | 0.82    | 0.76    | 0.58    | 0.51      | 0.54          |    0.63 |
> | Step-Audio      | 0.54    | 0.60    | 0.58    | 0.60    | 0.53      | 0.67          |    0.59 |
> | GPT-4o-Audio    | 0.51    | 0.48    | 0.52    | 0.53    | 0.38      | 0.44          |    0.48 |
>
>
> **Table C:** Inter-rater agreement across metrics/models between Model-as-Judge and human evaluation.
> | Model           | Text-C1 | Text-C2 | Text-C3 | Text-C4 | Audio-VES | Audio-Quality | Average |
> |-----------------|---------|---------|---------|---------|-----------|---------------|---------|
> | Qwen2.5-Omni-7B | 0.65    | 0.77    | 0.80    | 0.49    | 0.39      | 0.81          |    0.65 |
> | Step-Audio      | 0.66    | 0.51    | 0.45    | 0.55    | 0.45      | 0.72          |    0.56 |
> | GPT-4o-Audio    | 0.47    | 0.55    | 0.53    | 0.45    | 0.34      | 0.74          |    0.51 |
>
> ### Reference
> [1] Benchmarking Chinese Commonsense Reasoning of LLMs: From Chinese-Specifics to Reasoning-Memorization Correlations, ACL 2025
> [2] VoxDialogue: Can Spoken Dialogue Systems Understand Information Beyond Words? ICLR 2025
> [3] CosyVoice 2: Scalable Streaming Speech Synthesis with Large Language Models.

---

> ### Author Response · Authors · 2025-11-27
> **Revised Manuscript and Addressed Reviewer Concerns**
>
> Dear Reviewer zu4C,
>
> We have provided a detailed response to all of your questions and suggestions, and have uploaded a revised manuscript accordingly.
>
> **If our responses have addressed your concerns, we would kindly appreciate it if you could consider re-evaluating our score.**
>
> Thank you very much for your time and consideration.
>
> Best,
> Authors of Submission15900

---

### Official Review · Reviewer_zu4C · 2025-11-02

**Soundness:** 3
**Presentation:** 4
**Contribution:** 3
**Rating:** 8
**Confidence:** 3

**Summary:**

The authors propose EchoMind, a multi-level benchmark for evaluating empathetic capabilities of Speech Language Models, assessing whether models can perceive non-lexical vocal cues—beyond just spoken words—and respond with emotional intelligence.
The benchmark proposes 3 coarse-grained dimensions (speaker, paralinguistic, environmental) and 39 vocal attributes. It evaluates existing models with three tasks: understanding (content + voice perception), reasoning (integrated inference), and conversation (empathetic response generation), assessed via both objective metrics andModel-as-a-Judge/human evaluation. Testing 12 SOTA Speech LLMs reveals that while models excel at content understanding, they struggle with vocal-cue processing, and even state-of-the-art systems fail to generate emotionally aligned responses.

**Strengths:**

- Novel benchmark for evaluating empathetic capabilities of Speech LLMs. High quality taxonomy covering 39 attributes across speaker, paralinguistic, and environmental dimensions, providing comprehensive coverage of non-lexical vocal cues essential for human-like conversation.
- Rigorous evaluation in multiple setting, using both automatic and human evaluation at both text and audio levels. Moreover, it introduces specialized empathy metrics (EmoAlign, Vocal Empathy Score) and validates it on many state-of-the-art Speech LLM, revealing  limitations in vocal-cue integration and instruction-following.

**Weaknesses:**

Minor
- Majority TTS-generated data (646/1,137 scripts) with only 2 professional voice actors for human recordings, potentially missing natural variation and introducing artifacts that don't reflect real-world.

**Questions:**

URO-Bench and EChat-eval seems the most similar, could you elaborate more about the differences and the coverage?

---

> ### Author Response · Authors · 2025-11-21
> **Replies to Reviewer zu4C**
>
> Thank you for your time, careful review of our paper. Here are our detailed responses:
> >### **W1**. Concern on majority TTS-generated data with only 2 professional voice actors for human recordings, which may lack natural variation and real-world representativeness.
>
> We acknowledge that this limitation is objectively present and cannot be fully resolved in the short term.
> Current audio‑based benchmarks either utilize existing audio data for the target tasks[1][2][3] or, when such data are unavailable, typically synthesize speech using high-quality TTS models [4][5][6] such as CosyVoice2 [7] or GPT‑4o mini TTS [8].
>
> **In EchoMind, the synthetic portion was generated mainly via GPT‑4o mini TTS, with manual one-by-one synthesis to ensure high expressiveness and quality**. In addition, EchoMind includes a substantial human‑recorded subset (~43%) covering all 39 vocal attributes by two voice actors, enabling direct TTS–human comparison. This design ensures high consistency and controlled variation across attributes within limited resources. As shown in §4.3 (Figure 4), human speech poses greater challenges, further validating the need for this mixed data strategy.
>
> _**In Lines 229–233 of rebuttal revised manuscript, we emphasize the specific human verification procedures applied to TTS‑generated audio, and in Appendix A.3, we present additional details on human recordings.**_
>
> _______
> >### **Q1**. How does EchoMind differ from URO-Bench and EChat-eval?
>
> In terms of research focus, all three works target End-to-End Spoken Dialogue Models. The differences (as shown in Table 1 of the paper) are as follows:
> - At the **task type** level, EChat-eval focuses exclusively on the conversation task, whereas EchoMind and URO-Bench also encompass understanding and reasoning tasks. EchoMind’s audio data is purposefully constructed and shared across all tasks, allowing investigation into performance correlations between different task types. In contrast, URO-Bench’s audio data is derived from existing speech datasets or TTS-converted dialogue datasets, resulting in task data that is independent across tasks.
> - At the **audio attribute coverage** level, EchoMind includes 39 voice attributes specifically designed for dialogue scenarios, centered on the speaker and emphasizing emotional intelligence. EChat-eval contains only four attributes: emotion, age, gender, and sound event, while URO-Bench aims for comprehensiveness in spoken dialogue evaluation.
> - At the **voice attribute variation testing** level, EchoMind generates three distinct audio versions of the same semantically neutral dialogue script, each with different voice attributes, to examine whether voice characteristic differences elicit variations in model responses
> _______
> ### Reference
> [1] SD-Eval: A Benchmark Dataset for Spoken Dialogue Understanding Beyond Words, NeurIPS 2024
> [2] VoiceBench: Benchmarking LLM-Based Voice Assistants, 2024
> [3] AIR-Bench: Benchmarking Large Audio-Language Models via Generative Comprehension, ACL 2024
> [4] VoxEval: Benchmarking the Knowledge Understanding Capabilities of End-to-End Spoken Language Models, ACL2025
> [5] VoxDialogue: Can Spoken Dialogue Systems Understand Information Beyond Words? ICLR 2025
> [6] AudioBench: A Universal Benchmark for Audio Large Language Models, NAACL 2025
> [7] CosyVoice 2: Scalable Streaming Speech Synthesis with Large Language Models, 2025
> [8] https://platform.openai.com/docs/guides/text-to-speech

---

### Author Response · Authors · 2025-11-21

Dear Reviewers,

Thank you for your time, your thorough review, and your valuable comments and suggestions.

We have provided detailed responses and incorporated all relevant revisions into the rebuttal version of the manuscript.
We hope that our responses and revisions satisfactorily address your concerns and will be taken into consideration for a higher evaluation.

(Our response is lengthy due to the inclusion of the suggested fine‑grained analysis involving multiple SLMs, subtasks, and metric dimensions, resulting in large tables. Thank you for your understanding.)

Thanks,
Authors of Submission15900

---

### Meta-Review · Area_Chair_vK9r · 2026-01-07

**Summary:**

This paper proposes a new benchmark for evaluating empathetic dialogue responses in speech language models (SLMs). The benchmark
evaluates the level of spoken-content understanding, vocal-cue perception, reasoning, and response generation for 12 SLMs, and reveals that even the state-of-the-art SLMs struggle with expressive vocal cues, ans struggle with generating high quality empathetic responses.

**Reviewer Concerns:**

The following reviewer concern is not solved, and although the reviewer marks it as a minor concern, I think for speech research, this concern is nor minor: Majority TTS-generated data (646/1,137 scripts) with only 2 professional voice actors for human recordings, potentially missing natural variation and introducing artifacts that don't reflect real-world. An additional concern is that the two voice actors are non-native speakers, although they may be trained. I hope these issues could be fixed in the next iterations of the dataset.

**Reviewer Scores:**

I think none of the reviewers would change their scores based on the rebuttals.

---

### Decision · Program_Chairs · 2026-01-26

Accept (Poster)